# Association of Immune Thrombocytopenia and Inflammatory Bowel Disease in Children

**DOI:** 10.3390/jcm10091940

**Published:** 2021-04-30

**Authors:** Angela Guarina, Angelica Barone, Assunta Tornesello, Maddalena Marinoni, Giuseppe Lassandro, Paola Giordano, Milena Motta, Marco Spinelli, Ilaria Fontanili, Fiorina Giona, Francesco Menna, Elena Chiocca, Ilaria Fotzi, Angelamaria Petrone, Francesco Graziano, Paola Saracco, Giuseppe Puccio, Michele Citrano, Giovanna Russo, Piero Farruggia

**Affiliations:** 1UOC Oncoematologia Pediatrica—ARNAS Civico-Di Cristina-Benfratelli, 90127 Palermo, Italy; angelaguarina@gmail.com (A.G.); gipuccio@gmail.com (G.P.); pfarruggia@libero.it (P.F.); 2UO Pediatria e Oncoematologia—AOU, 43126 Parma, Italy; abarone@ao.pr.it (A.B.); ifontanili@ao.pr.it (I.F.); 3UOC Di Oncoematologia Pediatrica—PO “Vito Fazzi”, 73100 Lecce, Italy; assuntatornesello@gmail.com; 4SSD Oncoematologia Pediatrica-Ospedale Filippo Del Ponte, Varese ASST Settelaghi, 21100 Varese, Italy; maddalena.marinoni@asst-settelaghi.it; 5Department of Biomedical Sciences and Human Oncology, University of Bari “Aldo Moro”, 70121 Bari, Italy; giuseppelassandro@live.com (G.L.); paola.giordano@uniba.it (P.G.); 6UOC Ematologia ed Oncologia Pediatrica—AOU Policlinico “Rodolico-San Marco”, Università di Catania, 95131 Catania, Italy; milenamotta.mm@gmail.com; 7Fondazione MBBM/AO San Gerardo Clinica Pediatrica Universitaria, 20900 Monza, Italy; mspinelli@fondazionembbm.it; 8Policlinico Umberto I, Università La Sapienza, 00185 Roma, Italy; giona@bce.uniroma1.it; 9Dipartimento di Oncoematologia–AORN Santobono-Pausilipon, 80122 Napoli, Italy; franmen@tiscalinet.it; 10Oncologia, Ematologia e TCSE—Centro di Eccellenza di Oncologia ed Ematologia—AOU A. Meyer, 50139 Firenze, Italy; elena.chiocca@meyer.it (E.C.); ilaria.fotzi@meyer.it (I.F.); 11UOM Pediatria Ospedale S. Chiara, 38122 Trento, Italy; angelamaria.petrone@apss.tn.it; 12UO di Pediatria–Azienda Ospedaliera Ospedali Riuniti Villa Sofia-Cervello, 90146 Palermo, Italy; fra.graziano23@gmail.com (F.G.); citranomichele@gmail.com (M.C.); 13Dipartimento di Scienze Pediatriche e dell’Adolescenza, SC Pediatria Specialistica Universitaria, AOU Città della Salute e della Scienza, Presidio Ospedale Infantile Regina Margherita, 10126 Torino, Italy; paola.saracco@unito.it

**Keywords:** bowel, children, immune, thrombocytopenia, pediatric

## Abstract

Background: The association between inflammatory bowel disease (IBD) and immune thrombocytopenia (ITP) is still uncertain. In this multicenter retrospective study, the coexistence of both diseases was investigated in children diagnosed from 1 January 2000 to 31 December 2019. Methods: Clinical characteristics of both IBD and ITP, onset of disorders, and patient’s response to treatment were collected through a structured form sent to 55 Italian pediatric referring centers for hematological disorders. Result: Centers responded to the survey and reported the coexistence of IBD and ITP in 14 children. The first diagnosis was ITP in 57.1% and IBD in 35.7% of patients: it was simultaneous in 7.1%. IBD was classified as ulcerative colitis (57.1%), Crohn disease (35.7%), and unclassified (7.1%). No therapy for IBD other than steroids had any effect on ITP course. Colectomy resulted in recovery from ITP in 1 of the 2 patients surgically treated. ITP was always mild but turned to be chronic in half of patients. Conclusions: In all patients, ITP was mild without any evident impact on IBD severity, but the incidence of chronic ITP seems to be higher than what is usually observed in the pediatric age group. Colectomy had unpredictable effects on ITP.

## 1. Introduction

Many extraintestinal autoimmune manifestations are reported in inflammatory bowel disease (IBD), the most common hematological type being autoimmune hemolytic anemia [1,2,3,4]. In rare cases, immune thrombocytopenia (ITP), a hematological disorder characterized by the production of autoantibodies against platelets, can be associated [5,6,7,8,9,10,11]: only a few patients are of pediatric age [1,12]. In this retrospective study, we analyzed the largest series of children affected by both ITP and IBD.

## 2. Materials and Methods

This retrospective study was designed by the Coagulation Defects Study Group of A.I.E.O.P. (Associazione Italiana Emato-Oncologia Pediatrica). Informed consent for inclusion in the study was obtained from the parents or the legal guardians of all patients. A case report form (CRF) was sent to all 55 A.I.E.O.P. centers collecting information about IBD/ITP patients up to 18 years of age diagnosed for ITP or IBD or both from 1 January 2000 to 31 December 2019. Children affected by a known immunodeficiency were excluded. ITP clinical phenotype and IBD clinical phenotype + biopsy results were recorded. IBDs were classified as Crohn disease (CD), ulcerative colitis (UC) and inflammatory bowel disease unclassified (IBDU). ITP was defined as newly diagnosed (within 3 months from diagnosis), persistent (between 3 and 12 months from diagnosis), and chronic (cITP, lasting for more than 12 months) [13]. The severity of ITP was defined as [14]: score A, asymptomatic-paucisymptomatic ITP, clinical symptoms ranging from no bleeding to a few petechiae and some bruises without mucosal hemorrhages; score B, intermediate ITP, clinical picture with more petechiae, bruising, and mucosal hemorrhages; score C, severe ITP, clinical picture with severe cutaneous and mucosal bleeding symptoms with at least one of the following—retinal hemorrhages, intracranial hemorrhage, other severe internal hemorrhages, hemorrhagic shock, life-threatening bleeding. At the time of diagnosis, the patients were classified as: (1) IBD first (IBD was diagnosed before ITP); (2) ITP first (ITP was diagnosed before IBD); or (3) Simultaneous diagnosis (the second disease (IBD or ITP) was diagnosed during hospitalization or initial ascertainment for the other disorder). With regard to therapy efficacy, ITP complete response was defined as any platelet count ≥ 100,000/mmc with absence of bleeding; partial response was defined as any platelet count between 30 and 100,000/mmc with at least doubling of the baseline count and absence of bleeding; and no response was defined as any platelet count <30,000/mmc or less than doubling of the baseline count and/or occurrence of bleeding. Statistical analysis was performed using the open-source statistical software R (R Development Core Team 2011) [15].

## 3. Results

We collected CRFs regarding 14 IBD/ITP patients from 11 A.I.E.O.P. centers (Table 1): they were all Caucasian (M/F = 1/1). The most relevant clinical features of this cohort are shown in Table 2.

The median follow up (FUP) from ITP and IBD diagnosis was 6.2 (range: 0.05–15.8) and 3.7 years (range: 0.22–13.0), respectively. Median age at IBD and ITP diagnosis was 8.4 (range: 2.9–28.6) and 7.6 years (range: 2.4–15.8), respectively. A family history positive for other autoimmune disease was present in 4 patients (28.6%). Extraintestinal autoimmune disorders other than ITP affected 4 patients (28.6%): 2 out of 4 were psoriasis. The median number of platelets at ITP diagnosis was 31,500/mmc (range: 3000–98,000), and the severity score was A in 71.4% and B in 28.6% of patients, respectively. ITP turned to be persistent in 9/14 patients (64.3%) and chronic in 7/14 (50%): the median number of platelets at 3 and 12 months from ITP onset was 74,000 (range: 4000–366,000) and 100,000/mmc (range: 3000–500,000), respectively; 3 patients at 12 months from ITP diagnosis had platelets less than 20,000/mmc. At the last FUP 5/14 patients (35.8%) were presenting cITP, and 3 out of 5, still receiving treatment with eltrombopag, had acceptable platelet counts of 82,000, 91,000, and 93,000/mmc, respectively: these 3 children (Table 1) had been previously treated unsuccessfully with IVIG, steroids, MMF, and sirolimus (the first one), IVIG and steroids (the second one), and IVIG, steroids and MMF (the last one). Helicobacter pylori was tested in 10/14 patients using a stool antigen test and was found negative in all of them. Bone marrow aspiration was performed in 9/14 patients (64.3%) and was always consistent with ITP, showing normal to increased megakaryocyte numbers. The IBD was classified as UC in 57.1%, CD in 35.7%, and IBDU in 7.1%: Three out of fourteen patients had pancolitis. The first diagnosis was ITP in 8 patients (57.1%) and IBD in 5 patients (35.7%): it was simultaneous in 1 patient only. In the ITP first subgroup, the median time from diagnosis of ITP to diagnosis of IBD was 3.6 years (range: 0.8–12.8 years). In the IBD first subgroup, the median time from diagnosis of IBD to diagnosis of ITP was 2.1 years (range: 0.6–7.6 years). At ITP onset, 4/14 of the patients (28.6%) were not treated (wait-and-see), 8/14 (57.1%) received iv immunoglobulin (IVIG), and 2/14 (14.3%) oral prednisone. Subsequent rescue therapies for ITP were attempted in 7 patients (50.0%): most received IVIG or oral prednisone, 1 rituximab, 2 oral mycophenolate mofetil (MMF), 1 sirolimus, and 3 eltrombopag. No therapy for ITP, other than steroids, had any effect on IBD. Colectomy was performed in 2 patients: both presented complete remission from IBD and 1 of them also had a complete recovery from ITP.

## 4. Discussion

ITP is caused by antiplatelet autoantibodies that, binding to platelet membrane glycoproteins, mediate the destruction of platelets in the reticuloendothelial system [13,14,16]. The diagnosis of ITP is clinical, based on history, physical examination, and complete blood count with examination of peripheral blood smear and after exclusion of secondary causes, which are very uncommon in childhood. The association of ITP with other autoimmune diseases, above all thyroid autoimmune diseases and systemic lupus erythematosus, is well known [16,17].

IBDs represent a spectrum of enteropathies, including CD, UC, and IBDU, with a variable level of seriousness and a wide range of associated autoimmune disorders whose cumulative prevalence is 8.2% to 10.5% [18]. Autoimmune antibody-related hematologic diseases, such as autoimmune hemolytic anemia, autoimmune neutropenia, and ITP [5,6,19,20,21], have been occasionally reported in patients with IBD: up to now, fewer than 80 cases of IBD/ITP association of every age have been published, about two-thirds being UC [1,7,8]. There is a scarcity of literature about the association between IBD and ITP in childhood and, other than the cohort of 8 patients of Higuchi [12], to the best of our knowledge, only 15 other IBD/ITP pediatric patients have been published, mostly in single case reports [9,10,11,18,22,23,24,25,26,27,28,29,30]. Of these 23 pediatric patients, 12 were UC, 10 CD, and 1 IBDU. The present paper collected 14 patients, 8 of whom were UC; thus, in accordance with what was previously reported [12,23], it seems that UC is the type of IBD more commonly associated with ITP in childhood. Pancolitis was present in 3/8 UCs of our cohort (37.5%), a prevalence in line with what has been reported in childhood [31]. Evaluating the literature on the topic, it seems that, as far as the timing of the first diagnosis is concerned, the occurrence of ITP is much more common in childhood [12] than in adulthood [1,19,32]: our series validates this feature with more than half of the patients having presented ITP before IBD. This is an important point since, based on the much higher prevalence of patients with “IBD first” in adulthood, it has been frequently speculated that ITP in these patients is caused by IBD [6,19], secondary to immunostimulation from lumenal antigens: the pediatric experience, being the majority of the “ITP first” patients, seems not to corroborate this hypothesis, even though colectomy in 1 of the 2 surgically treated children resulted in complete recovery from ITP, and so in at least some patients a trigger mechanism from intestinal antigens cannot be excluded. One of the more interesting findings from our analysis is, in our opinion, the high incidence of cITP (50%), which is in accordance with the only one other cohort of pediatric patients [12], where 75% of children developed a chronic form. Therefore, it seems that in cases associated with IBD, the occurrence of cITP is higher than the 20–30% usually observed in the pediatric population [33], suggesting that a possible common underlying immune dysregulation can increase the rate of chronic forms, similar to what has been observed in secondary ITP in the course of other autoimmune disorders, such as lupus erythematosus or common variable immunodeficiency [34]. ITP features, irrespective of their appearance before or after IBD, are in our cohort very similar to what is typically observed in pediatric ITP, with mild onset and course: no patient suffered from severe hemorrhage and a rectal bleeding at IBD onset was experienced in 8 patients, 4 of whom were “ITP first” or “simultaneous” and 4 “IBD first”; so, it seems that a low platelet count does not affect in a definite manner the severity onset of enteropathy. In 2 out of 12 evaluable patients (16.6%), a concomitance between IBD flare and lowering of platelet counts was observed. From what has been previously published, no indication about the possible efficacy of any ITP therapy on IBD and vice versa can be drawn: analyzing our data, no IBD therapy other than steroids, in agreement to other previous reports [19], showed any clear impact on ITP and vice versa. Infliximab, a drug that showed conflicting but frequently good results on platelet count [6,29,35,36,37,38,39], was administered in 4 patients of our cohort but did not have any impact on ITP course. With regard to colectomy, whose efficacy on platelet count is often but not always found [1,12,22,26,27,32,40,41,42], it resulted in complete recovery from ITP in both of the 2 surgically treated patients of our series, but one of them exhibited an early ITP relapse only 4 weeks after colectomy: on this basis, in our opinion, it seems impossible to predict whether colectomy may have a definitive curative role in both ITP and IBD. Finally, we report 3 IBD/ITP children who, to the best of our knowledge, were the first to be treated with eltrombopag, a few of whom grew to adulthood [43]: all of them presented a sustained platelet response.

## 5. Conclusions

In summary, we present the largest case series reported to date of children affected by both IBD and ITP. With the inherent limitations of a retrospective design, we think that our results provide informative insights with regard to this rare pediatric population, where UC seems to be the most commonly associated type of IBD, the incidence of the chronic ITP form is higher than what is usually observed, colectomy has an unpredictable effect on ITP course, and steroids can be effective in both diseases. Finally, eltrombopag retains its efficiency in pediatric ITPs associated with IBD.

## Figures and Tables

**Table 1 jcm-10-01940-t001:** Characteristics of the patients.

Patient	Sex	Age at IBD Onset (Year)	IBD Type	IBD Therapies	IBD Status at Last FUP	Age at ITP Onset (Year)	ITP Therapies	cITP at Last FUP
1	M	7.58	CD	Steroid, AZT, adalimumab	Remission: under adalimumab	15.24	Wait-and-see	No
2	F	9.58	UC	Mesalazine	Mild disease: no therapy	11.69	IVIG, prednisone	Yes ^
3	F	19.05	UC	Infliximab	Remission: under infliximab	13.21	IVIG, prednisone, MMF, sirolimus, eltrombopag	Yes *
4	F	13.54	CD	Steroid, mesalazine, infliximab	Mild disease: under infliximab	9.97	IVIG, prednisone, MMF, rituximab	No
5	F	7.09	UC	Steroid, mesalazine, colectomy	Remission: no therapy	4.82	IVIG, prednisone	No
6	M	28.64	IBDU	Diet only	Remission: no therapy	15.85	IVIG, prednisone, eltrombopag	Yes *
7	F	9.23	CD	Steroid, metronidazole	Remission: no therapy	6.77	IVIG	No
8	M	6.64	UC	Steroid, mesalazine	Remission: under mesalazine	6.44	IVIG, prednisone	No
9	M	13.11	CD	Mesalazine	Remission: no therapy	12.30	IVIG	No
10	M	12.33	CD	Mesalazine	Remission: no therapy	2.45	IVIG	No
11	F	5.13	UC	Steroid, mesalazine	Remission: under mesalazine	11.99	Wait-and-see	Yes ^
12	F	2.93	UC	Steroid, mesalazine, AZT, CsA, thalidomide, infliximab, MTX	Active: under MTX and steroid	4.40	Wait-and-see	No
13	M	3.74	UC	Steroid, mesalazine, AZT, CsA, infliximab, colectomy	Remission: no therapy	4.43	IVIG, prednisone, MMF, eltrombopag	Yes *
14	M	12.46	UC	Mesalazine	Active: under mesalazine	8.85	Wait-and-see	No

IBD: inflammatory bowel disease; FUP: follow up; ITP: immune thrombocytopenia; CD: Crohn’s disease; AZT: Azathioprine; UC: ulcerative colitis; IVIG: intra venous immunoglobulin; MMF: Mycophenolate Mofetil; IBDU: inflammatory bowel disease unclassified; CsA: Cyclosporine A; MTX: Methotrexate. ^: not treated patients (wait&see); *: patients treated with eltrombopag.

**Table 2 jcm-10-01940-t002:** Characteristics of study population.

Characteristic	Value
Sex (F%)	50%
IBD diagnosis (years, median)	8.4
ITP diagnosis (years, median)	7.6
Platelets (median) at ITP diagnosis	31,500/mmc
Family history of autoimmune diseases	28.6%
Extraintestinal disorders	28.6%
IBD type	UC: 57.1% CD: 35.7% IBDU: 7.1%
ITP severity	Score A: 71.4% Score B: 28.6%
Timing of diagnosis	ITP first: 57.1% IBD first: 35.7% Simultaneous: 7.1%
Persistent ITP at 3 months	64.3%
Chronic ITP at 12 months	50.0%
Chronic ITP at last FUP	35.8%
First-line therapy for ITP	IVIG: 57.1%Wait-and-see: 28.6% Oral prednisone: 14.3%
FUP from ITP diagnosis (median, years)	6.2
FUP from IBD diagnosis (median, years)	3.7

IBD: inflammatory bowel disease; ITP: immune thrombocytopenia; CD: Crohn’s disease; UC: ulcerative colitis; IBDU: inflammatory bowel disease unclassified; FUP: follow up; IVIG: iv immunoglobulin.

## Data Availability

The data that support the findings of this study are available on request from the corresponding author. The data are not publicly available due to privacy or ethical restrictions.

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
