# Peer review of "Association of Immune Thrombocytopenia and Inflammatory Bowel Disease in Children"

_jcm, 2021, doi:10.3390/jcm10091940_

Round 1
Reviewer 1 Report
line 45: two patients are not enough to support this claim
line 85: I recommend the use of IBD/ITP instead ITP/IBD considering that ITP is probably a secondary ITP.
line 148/149: the data are not solid enough to support this claim
line 182/183: data supporting that eltrombopag retain its efficacy in pediatric ITP are discussed but not presented. Considerations upon treatment results would merit a more extensive discussion (or a separate table) within the Results chapter
Author Response
POINT 1, line 45: two patients are not enough to support this claim. ANSWER: the sentence “Colectomy can occasionally result in ITP recovery” was changed with “Colectomy has unpredictable effects on ITP”.
POINT 2, line 85: I recommend the use of IBD/ITP instead ITP/IBD considering that ITP is probably a secondary ITP. ANSWER: we made the substitutions.
POINT 3, line 148/149: the data are not solid enough to support this claim. ANSWER: the sentence “the pediatric experience, being the majority of patients “ITP first”, seems not to validate this hypothesis,” WAS CHANGED WITH “the pediatric experience, being the majority of patients “ITP first”, seems not to corroborate this hypothesis, even though colectomy in 1 of the 2 surgically treated children resulted in complete recovery from ITP, and so in at least some patients a trigger mechanism from intestinal antigens can not be excluded.”
POINT 4, line 182/183: data supporting that eltrombopag retain its efficacy in pediatric ITP are discussed but not presented. Considerations upon treatment results would merit a more extensive discussion (or a separate table) within the Results chapter. ANSWERS: as also asked by the other reviewer we added a table were the therapies at last FUP were added. Furthermore to the sentence “At the last FUP 5/14 patients (35.8%) were presenting cITP and 3 out of 5, still receiving treatment with eltrombopag, had acceptable platelet counts of 82,000, 91,000 and 93,000/mmc respectively:” WE ADDED: these 3 children (table 1) had previously been treated unsuccessfully with IVIG, steroids, MMF and sirolimus (the first one), IVIG and steroids (the second one) and IVIG, steroids and MMF (the last one).”
Reviewer 2 Report
49: Introduction.
The introduction completely lacks of bibliographic references on the presented topic and it must be expanded
67: The ITP bleeding score was defined as: this is a clinical classification of the severity of the disease rather than a Bleeding score (e.g. Buchanan and Adix)
84 Results.
There is no information describing the patients' features (age, sex, age of ITP / IBD onset, treatments for ITP / IBD, outcome) of each patient. A table summarizing these features could be helpful.
The incidence of ITP and IBD changes according to the age. In the results the median of the onset is similar in IBD and ITP. There are differences in the incidence of the two diseases in the different age groups?
It would be important to know the patients' inflammatory state. Are laboratory data available (e.g. PCR, ESR)? Did the therapies performed in each patient lead to a good control of the disease and changes in the inflammatory state?
168 Colectomy.
There is no information on the impact of colectomy on IBD in the 2 patients treated.
What is the hypothesis that explains the remission of ITP following colectomy?
Are there any laboratory data on the inflammatory status of the 2 patients before and after surgery?
Author Response
POINT 1. 49: Introduction. The introduction completely lacks of bibliographic references on the presented topic and it must be expanded. ANSWER: the references were modified and expanded.
POINT 2 The ITP bleeding score was defined as: this is a clinical classification of the severity of the disease rather than a Bleeding score (e.g. Buchanan and Adix). ANSWER. “ITP Bleeding score” was changed with “The severity of ITP”
POINT 3 There is no information describing the patients' features (age, sex, age of ITP / IBD onset, treatments for ITP / IBD, outcome) of each patient. A table summarizing these features could be helpful. ANSWER: the table (table 1) was added
POINT 4 The incidence of ITP and IBD changes according to the age. In the results the median of the onset is similar in IBD and ITP. There are differences in the incidence of the two diseases in the different age groups? ANSWER: we did not appreciate any difference.
POINT 5 It would be important to know the patients' inflammatory state. Are laboratory data available (e.g. PCR, ESR)? ANSWER; unfortunately these data are not available from our database.
POINT 6 Did the therapies performed in each patient lead to a good control of the disease and changes in the inflammatory state? ANSWER: these data are shown in the new table (table 1)
POINT 7 There is no information on the impact of colectomy on IBD in the 2 patients treated. ANSWER: It was added “Colectomy was performed in 2 patients: both presented complete remission from IBD e 1 of them had also a complete recovery from ITP”.
POINT 8 What is the hypothesis that explains the remission of ITP following colectomy? ANSWER: It was added the following sentence “…………even though colectomy in 1 of the 2 surgically treated children resulted in complete recovery from ITP, and so in at least some patients a trigger mechanism from intestinal antigens can not be excluded”
POINT 9 Are there any laboratory data on the inflammatory status of the 2 patients before and after surgery? ANSWER; unfortunately these data are not available from our database.